# The Corrosive Effects of Aftermarket Oil Additives on High-Leaded Tin Bronze Alloy

**DOI:** 10.3390/ma17061326

**Published:** 2024-03-13

**Authors:** Oriana Palma Calabokis, Yamid Nuñez de la Rosa, Paulo César Borges, Tiago Cousseau

**Affiliations:** 1GrMatS Group, Universidade Tecnológica Federal do Paraná, Curitiba 81280-340, Brazil; yenunezd@libertadores.edu.co (Y.N.d.l.R.);; 2Faculty of Engineering and Basic Sciences, Fundación Universitaria Los Libertadores, Bogotá 111221, Colombia; 3Centre for Bulk Solids and Particulate Technologies, The University of Newcastle, Shortland, NSW 2307, Australia

**Keywords:** copper alloys, corrosion, aftermarket additives, Cl-containing additives, S-containing additives

## Abstract

Aftermarket additives are used to enhance the performance of internal combustion engines in specific aspects such as reducing wear, increasing power, and improving fuel economy. Despite their advantages, they can sometimes cause corrosion-related problems. This research evaluated the corrosiveness of four aftermarket additives on the corrosion of a high-leaded tin bronze alloy over 28 days at 80 °C in immersion tests. Among the evaluated products, three showed corrosive effects ranging from intermediate to severe. Notably, the visual appearance of the surfaces often did not indicate the underlying corrosive damage. Therefore, the assessment of corrosiveness was based on chemical characterizations conducted on both the drained oils and the bronze surfaces. The study found minimal oil degradation under the testing conditions, indicating that the primary cause of corrosion was the interaction between the specific additives and the metal elements of the alloy, rather than oil degradation itself. A direct correlation was observed between the dissolution of lead and copper and the adsorption of S and Cl-containing additives on the surfaces, respectively. The corrosive impact of Cl-containing additives in aftermarket formulations was significantly reduced when mixed with engine oil SAE 10W-30 (at a 25:1 ratio), suggesting a mitigated effect in combined formulations, which is the recommended usage for engines.

## 1. Introduction

The internal combustion engine (ICE), integral to daily transportation and the automotive industry, converts only 21.5% of fuel energy into vehicle movement, as noted by Holmberg, Andersson, and Erdemir (2012) [1]. The remaining 78.5% of energy is lost: 33% to exhaust gases, 29% to the cooling system, and 38% to overcoming friction. This inefficiency underlines the continuous effort to improve ICE efficiency, a key area of technological and economic interest [1].

One strategy involves the creation of consumer-selectable additive packages, referred to as aftermarket oil additives. These additives serve multiple functions, primarily aiming to boost engine efficiency, fuel economy, and power, while also minimizing wear [2,3]. However, a significant concern, beyond the potential loss of the fully-formulated oil’s additive balance, concerns the possible side effects of corrosion, especially on the less noble metals of the ICE. Engine components, made from various metal alloys, are exposed to lubricant chemistry, with copper-based alloys being notably sensitive [4,5,6]. Consequently, standardized evaluation methods have been established, the most common being ASTM D130 [7]. The ASTM D130 method evaluates petroleum products corrosiveness by immersing a copper strip under specific conditions, classifying its effects via a color scale.

However, as Hunt (2018) [8] points out, the ASTM D130 test only provides a qualitative result without details on the reactions and chemical species involved, corrosion mechanisms, mass loss, or the formation of corrosion products. Consequently, various studies have evaluated the corrosiveness of lubricating oils or their additives, using complementary techniques for more comprehensive characterization [8,9,10,11,12,13]. Among the techniques employed, the following are notable: Inductively Coupled Plasma Atomic Emission Spectroscopy (ICP-AES) [11,12], Fourier-Transform Infrared Spectroscopy (FTIR) [13], Scanning Electron Microscopy (SEM) and Energy Dispersive X-ray Spectroscopy (EDS) [8,11], X-ray Photoelectron Spectroscopy (XPS) [10,12], Atomic Absorption Spectrometry (AAS) [9], and Total Acid Number (TAN) and Total Base Number (TBN) determination [9,12].

Kalantar and Levin (2008) [9] investigated the corrosion mechanisms of copper in the presence of oil degradation compounds. The authors suggest that partial oxidation products of the oil (such as hydroperoxides ROOH and free radicals RO•) can catalyze the dissolution of copper and/or the formation of CuO under aerated conditions (oven at 80 °C for 30 days). Jayne, Shanklin, and Stachew (2002) [12] found that the acidic products of the degradation of mineral base oil (Group I) without additives (at 135 °C for 216 h) were highly corrosive to lead (8000 ppm) compared to copper (negligible).

The study by Bruzzoniti et al. (2014) [14] elucidates how sulfur compounds affect copper’s reactivity, particularly in relation to temperature changes. Their study suggests that sulfur compounds with lower thermal stability or those more susceptible to oxidation tend to be less corrosive at higher temperatures, as the processes of thermal or oxidative degradation mitigate the corrosive reactions to copper [14]. In a related study, Reid and Smith (2014) [10] used XPS to analyze the surfaces of pure copper strips tested in iso-octane with elemental sulfur and sulfur compounds (immersion test at 50 °C for 3 h). They concluded that only elemental sulfur caused corrosion according to the ASTM D130 color standard.

Jayne, Shanklin, and Stachew (2002) [12] explored the corrosive effects of various additives in oil on copper and lead by conducting immersion tests (at 135 °C for 216 h). Their oil formula included dispersants, ZDDP (Zinc Dialkyl Dithio Phosphate), detergents, antioxidants, and antifoaming agents in mineral base oil. They found the oil more corrosive to lead (100–150 ppm) than copper (10–20 ppm), linking this to the corrosiveness of secondary and primary ZDDP respectively. Increments in amine dispersant content and the TBN of the dispersant significantly influenced corrosion, resulting in pronounced dissolution of lead (Pb) while reducing corrosion for copper (Cu). The study highlighted the critical role of ZDDP types and the synergistic effect of dispersants in mitigating corrosion. Specifically, it was confirmed by XPS, that copper sulfide formation leads to copper corrosion, whereas lead sulfides protect against lead corrosion [12].

As observed, the majority of research has primarily focused on studying the corrosiveness in pure copper. However, the galvanic effects of other elements in a copper-based alloys are also significant. Hunt (2018) [8] noted that corrosion measurements in pure copper do not accurately indicate the corrosion behavior of brass when exposed to commercial transmission lubricants. Bares et al. (2020) [11] investigated the impact of brass composition and oil formula, finding that a Cu_63_Zn_37_ alloy with automotive gear oil (AGO) formed corrosive deposits between 80 and 120 °C, whereas a Cu_85_Zn_15_ alloy only started corroding above 110 °C. These differences are attributed to the complex interactions between the selective dissolution of Zn and Cu, and the formation of surface films containing sulfur, phosphorus, and oxygen [11].

Previous research indicates that analyzing the corrosiveness of oils and their compounds is a multi-component and complex issue. To the best of the authors’ knowledge, there have been no studies focused on the corrosiveness of commercial aftermarket additives to date. Consequently, this study investigates the corrosive effects of four commercial aftermarket additives, specifically metal conditioners [15,16], on a bronze alloy. Utilizing a comprehensive approach, the research employs surface characterization techniques, including optical microscopy, FTIR, and SEM-EDS. Additionally, the study examines degradation, metal content, and additive consumption in drained oils using FTIR and ICP-OES, and detects chlorine and sulfur in additives with Energy-Dispersive X-ray Fluorescence Spectrometry (EDXRF). Together, these complementary techniques enabled the determination of the corrosiveness level of each additive.

## 2. Materials and Methods

### 2.1. Materials

The samples were metal plates made of high-leaded tin bronze alloy, known as SAE TM 23, with dimensions of 30 × 10 × 2 mm. The chemical composition of the material, as provided by the supplier and determined by SEM-EDS, is presented in Table 1. The microstructure consists of islands or globules of lead within a matrix of copper, zinc, and tin, which are the principal elements of the alloy. This chemical composition and microstructure are responsible for its properties, such as high corrosion resistance (among bronze alloys), excellent anti-friction quality, optimal machinability, and formability. Owing to these properties, high-leaded tin bronze alloys are preferred materials for manufacturing various components such as valves, bushings, bearings, sleeves, crowns, rings, and hydraulic materials.

The bronze pieces were sanded with SiC papers up to a P1200 grit size. They were then cleaned in an ultrasonic bath with pure acetone and dried with warm air. Each sample was placed in a separate glass container (test tubes), containing 3 mL of lubricant. The containers were cleaned in the same way as the samples. The test lubricants consist of commercially available aftermarket additives in the Brazilian market, known as metal conditioners (MCs) [15,16]. Additionally, a mixture of a fully formulated SAE 10W-30 API SL JASO MA (motorcycle engine oil) with Metal Conditioner D (MC-D) was tested. The proportion recommended by the MC manufacturers was used: 25:1 mL. MC-D was chosen because it exhibited the highest corrosiveness. The properties of the evaluated lubricants are listed in Table 2.

### 2.2. Procedure

The glass containers containing the samples (both bronze and lubricant) were hermetically sealed and placed in a heating oven, maintained at a controlled temperature of 80 ± 1 °C. Control samples consisting solely of the lubricants (without any immersed parts) were also prepared. The bronze parts were positioned vertically, immersed up to about a third of their height (~10 mm), as depicted in Figure 1. Containers with each type of tested lubricant were removed at intervals of 14, 21, and 28 days. Subsequently, the bronze pieces were extracted, cleaned in an ultrasonic bath using pure acetone, dried with warm air, and stored in a vacuum to prevent oxidation prior to the surface characterization.

### 2.3. Analysis of Lubricants

Lubricant samples from 0 (fresh oil), 14, 21, and 28 days were characterized using FTIR (Shimadzu, model IRAffinity-1) with an Attenuated Total Reflection (ATR, ZnSe crystal) in accordance with ASTM E2412 [17]. The measurement resolution was set at 4 cm^−1^, and the scanning range spanned from 4000 to 650 cm^−1^. Sixty-four scans were conducted for each lubricant sample. The aim of this analysis was to monitor the degradation of the lubricant. The acidity of both new and used lubricants was measured using pH paper strips. Utilizing this method, no changes in the acidity of the lubricants were recorded over the course of the test.

The elemental composition of the oils before and after immersion were monitored using Inductively Coupled Plasma Optical Emission Spectrometry (ICP-OES, PerkinElmer, Optima 7300 DV). These analyses were carried out by a laboratory specialized in lubricant analysis, which follows the procedures of the ASTM D5185 standard [18]: the oil sample was diluted ten times by weight with analytical grade solvent using an ultrasonic homogenizer: isoparaffin in most cases, and xylene for those that were difficult to be correctly homogenized in isoparaffin.

However, ICP-OES does not allow for the determination of chlorine at any concentration and sulfur only within the range of 900–6000 ppm (parts per million, Ref. [18]). As a result, the presence of chlorine and sulfur in the lubricants listed in Table 2 was qualitatively determined using Energy-Dispersive X-ray Fluorescence Spectrometry (EDXRF). The spectra were acquired using a spectrometer (model EDX-720, Shimadzu, FAPESP Project 2009/08584-6), which employs a Rh tube, a voltage of 15–50 kV, variable current of 1–1000 µA, air atmosphere, and a Si(Li) semiconductor detector cooled to the temperature of liquid nitrogen. The samples were placed in polyethylene sample holders and positioned over a special thin polystyrene film (Mylar^®^). Each sample used 200 µL, filling the 7 mm bottom of the sample holder. No extraction or digestion process was applied to the oils. A qualitative determination was performed, obtaining the values of relative intensities corresponding to the characteristic fluorescence spectra of chlorine (Cl-Kα) and sulfur (S-Kα).

### 2.4. Analysis of Metal Surfaces

The adsorbed additives on the surfaces of the metal parts were analyzed using FTIR-ATR. A spectrophotometer (Shimadzu, IRSpirit-T, Single Reflection ATR accessory) was used, with a resolution of 4 cm^−1^ and a reading range from 4000 to 650 cm^−1^. Spectra were acquired in the immersed region, at the interface, and in the external region (not in contact with lubricant) of the bronze pieces (regions indicated in Figure 1).

The appearance of the bronze surfaces was monitored using microscopy techniques at 14, 21, and 28 days. Macrographs were taken with a stereo microscope (Olympus SZX10), while micrographs were acquired using an optical microscope (Olympus BX51RF). Additionally, micrographs with higher magnifications and semi-quantitative chemical compositions were taken using a Scanning Electron Microscope equipped with Energy Dispersive X-ray Spectroscopy (SEM-EDS; SEM: Zeiss EVO MA15/EDS: Oxford Instruments X-MAX 20). Before SEM-EDS characterization of bronze surfaces, the pieces were removed from the vacuum and cleaned again in an ultrasonic bath with pure acetone.

The SEM-EDS semi-quantitative analyses were conducted using the standardless analysis method, as recommended by the equipment manufacturer. Measurement parameters were kept constant for all samples. A beam measurement on pure copper was performed before analyzing each bronze sample. To calculate the chemical composition, the pre-installed standardization database was used. The elements in the spectra were automatically identified, and the results were presented as the percentage by weight (wt.%). All elements identified through EDS mapping (performed at a magnification of 400×) were taken into account, with the exception of carbon, which was excluded due to its classification as a contaminant (base oil residue). The composition was presented as the average of six maps obtained from both the inside and outside regions (Figure 1).

## 3. Results

### 3.1. Analysis of Bronze Samples

Figure 2 displays images of the pieces in their end-of-test condition at 14, 21, and 28 days. Approximately one-third of each piece’s surface, specifically the lower region shown in Figure 2, was submerged (called inside region in Figure 1). All samples exhibited color changes during the test, varying in intensity and location. The color change was particularly noticeable in the MC-A samples (Figure 2a,f,k) and the mixture (Figure 2e,j,o) in the non-submerged areas, whereas the MC-B sample (Figure 2b,g,l) showed color changes throughout the entire piece. Microscopic observations do not conclusively determine whether there was corrosion in the MC-A, MC-B, and mixture samples throughout the duration of the test.

On the other hand, the samples tested in MC-D showed corrosion at the interface between the lubricant and the air from 14 days (Figure 2d), which then propagated until the end of the test from the interface (Figure 2n). Regarding the MC-C samples, corrosion began at the lower end after 21 days (Figure 2h, marked by a dashed yellow square) and by the end of the test it had extended throughout the entire submerged region (Figure 2m).

Higher magnification optical micrographs within the dashed squares of Figure 2 were taken to detail the corrosion of the samples tested in MC-C and MC-D. Figure 3 shows the magnified views of these dashed regions for all evaluation periods. On the surfaces in contact with MC-C (Figure 3a–d), an evolution of corrosion products was observed throughout the testing period. This progression appears to have begun at 21 days at the lower end (Figure 2h and Figure 3b) and subsequently spread uniformly across the immersed region (Figure 2m and Figure 3c,d). The surface texture of the piece tested in MC-C (Figure 3b–d) seems to be the result of the growth of a layer of corrosive deposits emerging from the surface.

In MC-D, the evolution of corrosion followed a different mechanism, beginning and spreading from the interface between the lubricant and air (Figure 2d). In the magnified view of this interface in Figure 3e (14 days), regions of the sanded surface that remained unattacked are observable, while the extensive matrix underwent corrosion (dark areas). At 21 days, at the interface (Figure 3f), regions of the sanded surface are still visible along with the growth of a layer of corrosion products projecting from the surface. By the end of the test, the corrosion morphology differed between the interface (yellow square in Figure 2n) and the immersed region (blue square in Figure 2n). At the interface (Figure 3g), the surface texture results from corrosion products with varying shades. In the immersed region (Figure 3h), the original sanded surface marks are still visible alongside areas of localized attack as well as widespread surface corrosion.

The described differences between the pieces tested in MC-C and MC-D suggest that certain additives in the formulations are corrosive to the bronze alloy. These additives are likely of different chemical nature, as they act through distinct corrosion mechanisms. Based on the images in Figure 2e,j,o, it is important to highlight that such additives from MC-D, when mixed with the commercial oil 10W-30, did not produce apparent corrosion.

### 3.2. Analysis of Lubricants

Since oxidation and corrosion are inseparable phenomena, the drained lubricants were analyzed using FTIR to assess their degradation and ICP-OES to quantify corrosive wear metals and additive consumption. Table 3 shows the variation between the new and tested products in the region associated with oxidation (1800–1670 cm^−1^) according to ASTM E2412 (FTIR, Ref. [17]). Figure 4a representatively presents the results of one of the conditioners in the oxidation region, as the others showed similar trends.

Given the short test period (28 days) and the absence of tribological contact, a proportional evolution of oxidation over time was not observed (Table 3). This fact is also associated with the presence of compounds with C=O bonds verified in the spectra of all new oils, as seen in Figure 4a for MC-B. These compounds may come from an ester base oil or ester-based additives. Therefore, the oxidation analysis should be comparative to the new oil rather than absolute, as is usually enacted for engine oils. Thus, Figure 4a shows a decrease in the intensity of the major C=O bond peak (~1740 cm^−1^) in the tested oils compared to the new oil. This decrease occurred in all tested lubricants and is associated with the reduction in additives containing the C=O bond, either due to their migration to the piece’s surface (adsorption) or their thermal degradation. On the other hand, during the test, an increase in absorbance at the peak of ~1712 cm^−1^ compared to the new condition (Figure 4a) was noted, possibly due to mild oil oxidation, increasing the measured area in the region between 1800 and 1670 cm^−1^ (Table 3).

Calabokis et al. (2022) [15] and Macián et al. (2012) [19] also observed variations in oxidation values that did not align with the degradation time of used oils. They attributed these changes to the presence of esters, both in the additive load [15] and in the formulation of ester-based oils [19]. Specifically, Calabokis et al. (2022) reported percentage changes in oxidation in used motorcycle crankcase oils containing metal conditioners, ranging from −23% to +220%, with an average oxidation increase of 27% over three oil changes [15]. In the tests presented here, the maximum variation associated with the consumption or degradation of C=O compounds in the mixture was −6.41% (new vs. 14D), and the maximum variation due to oil oxidation was +4.36% (new vs. 28D). This indicates that the test results (Table 3) are consistent with the values observed in real-world conditions.

Additionally, for all lubricants, an increase in absorbance was observed in the region between 3600 and 3150 cm^−1^ (Figure 4b, Table 3), compared to the condition of new (fresh) oil. Figure 4b representatively shows the results for one of the conditioners, as similar trends were observed in the others. The 3600–3150 cm^−1^ region corresponds to the intermolecular O-H bond [17]. Therefore, an increase in this region is commonly observed due to water contamination [17] or the formation of degradation products in the oil, such as alcohols, carboxylic acids, and OH- radicals [9]. Considering that the tests were conducted under ‘closed-cap’ conditions (Figure 1) with limited air exposure, and that the spectra of lubricants in contact with the parts (over all time periods) showed higher absorbances than the 28-day control oil in the 3600–3150 cm^−1^ region (Table 3), it is likely that the observed increase predominantly relates to the formation of oil degradation products. Some of these products may have originated from the decomposition of ester-type additives, as indicated by the decreased absorbance at the ~1740 cm^−1^ peak observed in Figure 4a, a trend observed across all lubricants.

Table 4 displays the concentrations in parts per million (ppm) of corrosive wear metals (Cu, Pb, and Zn) and the elements from the additive load, as determined by ICP-OES. The analysis revealed zero (0) ppm for other components of the alloy: Fe, Ni, and Sn. 

Table 4 reveals that there was a detachment of copper in the tests conducted with MC-A, MC-C, and MC-D. In the case of MC-A, the values were only 1 ppm higher than the new oil, while in MC-C and MC-D oils, increases of 3.3 and 32.5 ppm were observed, respectively. For MC-C, copper was detected only at 28 days, which aligns with the visual observations presented in Figure 2m, where extensive corrosion in the immersed region is only observed on day 28. MC-D caused corrosive wear with detachment of Cu and Zn particles at all evaluation intervals. The amount of Cu in MC-C and MC-D after 28 days seems small compared to the extensive corrosion they caused in the bronze parts (Figure 2 and Figure 3). This is due to the limitations of ICP-OES spectroscopy, whose results are sensitive to particle size [18,20]. Indeed, in the oils drained from MC-C and MC-D after 28 days, clusters of corrosion products of millimeter sizes were observed.

An unexpected result was the Pb content in the drained MC-A and the mixture (Table 4). This reveals that they caused selective corrosion of lead in the high-leaded tin bronze parts during all evaluation periods. In these lubricants, the Pb content increased over the duration of the test. These results are surprising since neither the macrographs in Figure 2 nor the micrographs exhibit evident signs of corrosion. The only notable change was the loss of color and brightness in the immersed region of the pieces tested with MC-A (Figure 2a,f,k) and the mixture(Figure 2e,j,o).

Table 4 also presents the ICP-OES results for some elements of the additive package (P, Ca, Mg, B, and Na). A general trend observed for all lubricants is the reduction in the additive load compared to the new oil. This reduction was always greater for the 14D condition compared to the 21D and 28D conditions. This result is associated with the adsorption of additives on the virgin metal surface, leading to their depletion in the oil. Over time, due to the oxidation of the surfaces, these additives probably return to the oil. Additionally, the 28D-control condition has a lower content of elements compared to the new condition, a result that could be linked to the thermal degradation of the additives or the lubricant, as observed in field tests on vehicles by Agocs et al. [20]. Specifically, they noted that the presence of oil degradation products accelerated the depletion of additives, especially antioxidants [20].

The analysis of Zn is more complex since it is found both in the lubricating oil (additive package) and in the tested bronze alloy (Table 1). Consequently, the presence of Zn in the used oils could originate from either or both sources. The observed reduction in Zn content—relative to its initial state—in tests involving MC-A, MC-B, and the mixture, correlates with the consumption or degradation of Zn-containing additives over time, as was observed for the other additive elements (P, Ca, B, and Na). In the case of MC-C, the formulation does not include Zn, which is consistent with its absence in the drained oil. Conversely, despite also not including Zn in its formulation, the drained MC-D exhibited Zn concentrations beginning at 21 days, suggesting a process of selective corrosion of Zn.

The presence of S- and Cl-containing additives in the formulation of the metal conditioners is expected, given their role in extreme pressure and anti-wear functions. In this context, the presence of chlorine and sulfur in new oils was qualitatively assessed using EDXRF. Characteristic fluorescence spectra for this analysis are displayed in Figure 5, but only for the samples MC-B, MC-D, and SAE 10W-30 API SL JASO MA. The S-Kα and Cl-Kα were used as analysis lines, highlighted in yellow (Figure 5). The red curves denote the background interference from scattered X-rays originating from the X-ray tube, which in this case employed a Rhodium (Rh) target. The blue curves in Figure 5 correspond to the sample’s spectra.

Notably, the spectra of MC-B (Figure 5a), MC-A (not included in the figure), and SAE 10W-30 (Figure 5c) revealed a distinct peak in the S-Kα region. Whereas for MC-C and MC-D (Figure 5b), no prominent signal appeared in the S-Kα region but rather in the Cl-Kα region. Additionally, the SAE 10W-30 sample (Figure 5c) also exhibited a peak for Cl-Kα. However, the analysis of chlorine posed a challenge due to background interference. The characteristic X-rays from the X-ray tube overlapped with the target peak in the spectra of MC-A and MC-B (Figure 5a), particularly in the range of 2.5–3.0 keV. This overlap complicates the confirmation of chlorine compounds in these samples.

According to [21], matrix effects significantly impact the quantitative determination of sulfur in petroleum-derived products when using EDXRF. Such effects arise from the fluid’s composition, like the oxygen content or the carbon/hydrogen ratio [21]. In this study, no corrective procedures to account for matrix effects or calibration curves were implemented. Therefore, the quantitative data provided by the equipment (expressed as intensity in cps/µA) that is directly related to the content of that element, should be interpreted only for comparative ranking purposes, given that all samples were analyzed under the same measurement conditions. The intensity results (cps/µA) for the S-Kα line were as follows: 10.1053 (MC-B) > 2.6267 (MC-A) > 0.6969 (SAE 10W-30) > 0.2292 (MC-D) > 0.1868 (MC-C). Regarding the Cl-Kα line, the intensities were ranked as follows: 24.354 (MC-D) > 15.2156 (MC-C) > 0.3031 (SAE 10W-30) > 0.2176 (MC-A) > 0.1086 (MC-B). Based on the mentioned considerations and the obtained results, it is only possible to affirm the presence of sulfur in the MC-A and MC-B additives, and in the SAE 10W-30 oil. The presence of chlorine is confirmed in the MC-C and MC-D, and also, in a lower amount, in the SAE 10W-30 oil. These results are crucial for the analyses of the surfaces presented in the following section.

In the analysis of the drained oils, a slight degradation was observed by FTIR, which is expected considering the short testing period (28 days) and the absence of tribological contact. All lubricants showed a reduction in the intensity of the ester peak and an increase in the absorption of the intermolecular O-H bond. These changes are indicative of the consumption of additives containing the C=O bond and overall lubricant degradation. Moreover, the level of additive elements in the drained oils showed a decrease across all testing periods compared to their initial conditions. These reductions can be attributed to thermal degradation or adsorption onto surfaces. On the other hand, the detection of metals from the bronze alloy, such as copper, zinc, and lead, in the drained oils points to the corrosive effect of some products on these specific metals.

### 3.3. Analysis of Metal Surfaces

The chemical composition of the bronze surfaces was analyzed using FTIR and SEM-EDS. It is crucial to highlight that these characterizations were performed following the cleaning of the samples. As a result, the findings exclusively represent the corrosion products and additives that remained adsorbed onto the surfaces.

Figure 6 presents the FTIR spectra of the surfaces. For comparative purposes, the spectrum of each oil in its new condition is also presented, with the analysis conducted in the liquid state. Overall, Figure 6 demonstrates that a limited number of organic groups remained adsorbed on the metal surfaces after the cleaning procedure. The characteristic peaks of the surface spectra are broader and less intense compared to the spectrum of the new oils, this observation aligns with the findings of Piras, Rossi, and Spencer (2003) [13].

In the case of MC-A and MC-B (Figure 6a), results are only shown for the end of the test in the immersed region, which show absorbance attributable to the aliphatic groups of hydrocarbons (νCH_2_). For other evaluation times, the spectra did not exhibit any absorbance for the parts in MC-A and MC-B, and hence are not presented. For the MC-C (Figure 6b), MC-D (Figure 6c), and the mixture (Figure 6d), the spectra are provided for all evaluation times, both in the immersed region and at the interface. These spectra clearly demonstrate the adsorption of various organic compounds, including the stretching and bending vibrations of C-H, such as νCH_2_, δCH_2_, and δCH_3_. Interestingly, the adsorption of the carbonyl group (C=O) on the surfaces was only confirmed for the MC-C samples (Figure 6b), despite all the new additives having this characteristic peak in their spectra.

In the spectra of the parts tested in MC-C (Figure 6b), MC-D (Figure 6c) and in the mixture (Figure 6d), peaks identified in the region between 1000 and 775 cm^−1^ were associated with stretching vibrations of P-O-P and P-O-C (νP-O-P, νP-O-C). According to [13], these bonds originate from the degradation of antiwear additives such as Zinc dialkyldithiophosphates (ZDTPs). Furthermore, it is noteworthy that from 21 days onwards in MC-C and MC-D, the peak located at 585 cm^−1^ could possibly be associated with the stretching of halo compounds such as P-Cl or C-Cl. This is supported by the detection of chlorine by EDXRF and in the surface analyses performed by SEM-EDS, as will be presented subsequently.

Finally, one or two broad bands in the 3500–3250 cm^−1^ region associated with the O-H stretching vibration (νO-H) are identified in the spectra of the surfaces in MC-A, MC-C, MC-D, and the mixture (Figure 6). The presence of the O-H group on the surfaces confirms the hypothesis raised regarding the spectra of the drained oils (Figure 4b and Table 3): partially degraded oil products have formed and remain adsorbed on the bronze pieces.

The results of the semi-quantitative elemental chemical composition analyzed by SEM-EDS are displayed in Table 5. Considering the necessity for rigorous sample cleaning in SEM-EDS analysis, the following results are confined to examining the elemental composition of compounds formed between the additives and the metallic matrix, which remained adsorbed despite ultrasonic cleaning. The EDS analyses of the surfaces tested in MC-A and MC-B (Table 5) corroborate the results shown in Figure 6a, indicating limited adsorption of additives. Chlorine was detected on the surfaces analyzed with MC-A and MC-B, displaying low concentrations and a wide range of results. Given that its determination is semi-quantitative, such results indicate that its presence is due to contamination rather than being an additive, which is corroborated by the findings obtained through EDXRF (Figure 5). Specifically, in the surfaces with MC-A, the presence of sulfur was confirmed across the entire piece. In the pieces with MC-B, calcium was identified in the immersed region.

The oxygen content of the surfaces before testing was 3.52 wt.%. Thus, the results in Table 5 suggest that the formation of surface oxides in MC-A and MC-B was minimal compared to the high oxygen content observed in conditions MC-C, MC-D, and the mixture. Notably, Table 5 presents results in the interface region for MC-D, as this was the only condition that exhibited preferential corrosion in this area (refer to Figure 2d,i,n). Chlorine was the sole additive element detected on the surfaces tested in MC-C and MC-D. The surface region with the highest chlorine content appears to coincide with the incidence of corrosion: a more significant attack was observed with MC-C in the immersed region (Figure 2m and Figure 3d) and with MC-D at the interface (Figure 2n and Figure 3g). The parts tested in the mixture showed the greatest diversity of additive elements (Ca, Na, Cl). This outcome is a result of the lubricant used, SAE 10W-30, being a fully formulated oil.

The analyses conducted using SEM-EDS enabled the identification of a preferential distribution of the additive elements sulfur (S) and chlorine (Cl) in the microstructure of the bronze, as demonstrated in Figure 7. Under the conditions in which sulfur or chlorine are adsorbed onto immersed surfaces (Table 5), a representative case for each of these elements was selected for presentation in Figure 7a4,b4, respectively.

In Figure 7a, a significant correlation emerges between the intensity of the sulfur element (Figure 7a4) and the presence of lead islands (Figure 7a3) within the microstructure. Conversely, the analysis presents a different scenario for chlorine, as depicted in Figure 7b. The map for chlorine (Figure 7b4) displays extensive coverage across the matrix of the microstructure, predominantly composed of copper. These findings indicate a selective affinity: sulfur for regions rich in lead, and chlorine for the copper matrix. This pattern is likely attributable to the formation of sulfide and chloride compounds, respectively, in these specific areas.

## 4. Discussion

### 4.1. Corrosiveness Ranking

In this study, we observed color changes in the bronze components of all evaluated products, with variations in both intensity and location (refer to Figure 2). Through comprehensive analyses using techniques such as ICP-OES and FTIR on drained oils, and FTIR and SEM-EDS on bronze surfaces, these color variations were identified as signs of corrosion, ranging from mild to severe. Among all the lubricants evaluated, MC-B was the only lubricant without corrosive effects under the conditions tested in this research. The remaining products were ranked according to their corrosivity, in ascending order: mixture = MC-A < MC-C < MC-D. This ranking supports the findings of Kalantar and Levin (2008) [9]. They advocate for a broader focus on factors like the copper content in the oil or the Total Acid Number (TAN), rather than limiting evaluations to color changes as proposed by the ASTM D130 standard.

The bronze pieces immersed in MC-B only showed coloration changes (as shown in Figure 2b,g,l) and no alloy metals were detected in the drained oil (Table 4). Also, the low levels of oxygen on the surfaces, indicates a limited formation of oxides. One potential explanation is that the drained oils indeed contained alloy elements, yet their concentrations fell below the detection limit of the ICP-OES technique, suggesting a very slight degree of corrosion. On the other hand, it is also possible that its additives (detergent, antioxidants, or corrosion inhibitors, associated with the contents of Zn, P, Ca, Mg, and Na in the new MC-B, as per Table 4) played a key role in surface protection.

The test results with MC-A and the mixture can be categorized as a mild corrosive attack: the bronze surfaces only exhibited loss of coloration (as seen in Figure 2) and the ICP-OES analyses (Table 4) suggest a selective corrosive attack on lead, whose content progressively increased over time in the drained oils. In comparison to the tests with MC-A, the mixture led to the formation of a layer of surface oxides, as confirmed by the higher oxygen content (up to 16.43 wt.%, Table 5). Additionally, the presence of sulfur on the surfaces (Table 5) and in the oil formulations, can be linked to this localized attack, as seen in Figure 7a. Consequently, in both cases the whitening of the surface in the immersed region and the interface is related to the depletion of lead from the surfaces.

Finally, the MC-C and MC-D additives demonstrated a severe corrosive effect on the bronze alloy under evaluation. Both exhibited signs of obvious macroscopic corrosion (as shown in Figure 2), along with the presence of copper and zinc contents (Table 4) and millimetric particles of corrosive wear in the drained oils. Additionally, throughout the test, a layer of corrosion products projecting from the surface was observed to grow (as seen in Figure 3) in the regions most affected by each lubricant. The SEM-EDS analyses (Table 5) suggest that the area with the highest concentration of chlorine coincides with the incidence of corrosion: greater attack was observed with MC-C in the immersed region (Figure 2m and Figure 3d) and with MC-D at the interface (Figure 2n and Figure 3g).

Despite FTIR determinations that organic groups such as P-O-P, P-O-C, P-Cl, or C-Cl were adsorbed on the surfaces immersed in MC-C and MC-D (Figure 6b,c), only those containing chlorine were detected by SEM-EDS (Table 5, Figure 7b). Two hypotheses can be considered: (i) Due to the test conditions (80 °C, no tribological contact), only chlorinated compounds remained adsorbed on the surfaces after the cleaning procedure (before SEM-EDS analysis); (ii) the P-containing compounds were not present in sufficient quantities to be detected by SEM-EDS. The greater corrosive power is attributed to MC-D, which initiated corrosion in a shorter time (14 days, as seen in Figure 2d) compared to MC-C (21 days, as shown in Figure 2h). This analysis underscores the complex interactions between lubricant additives and metal surfaces, highlighting the importance of considering the specific chemical nature of additives and their potential for causing corrosion, which will be discussed in the subsequent section.

### 4.2. Corrosiveness and Oil Additives Relationship

The adsorption of lubricant molecules on metal surfaces occurs primarily through two mechanisms: physical and chemical [22,23]. The former, otherwise known as physisorption, takes place when the interaction between the adsorbed molecule and the surface is weak, resulting from secondary chemical bonds. In contrast, chemical adsorption, or chemisorption, results from primary chemical bonds such as ionic or covalent bonds. Due to the temperature of the oil bath (80 °C) and the absence of tribological contact, it is expected that only the physical adsorption of additives onto the virgin surfaces occurred [12,13,22,23]. This would explain why only a few additive elements (SEM-EDS: Table 5) and organic groups (FTIR: Figure 6) were detected on the surfaces of the parts, despite the diverse chemical composition of the formulations (Table 4, Figure 5). Also, another possibility is that some elements were not present in sufficient quantities to be detected by SEM-EDS or they were removed by the cleaning process. Based on the additives and functional groups that remained adsorbed on the surfaces and the characterizations of the drained oils, the mechanisms of corrosion involved will be discussed.

The corrosive effects of lubricating additives have primarily been studied on pure copper [3,5,6,8,9,10,12,14], as suggested by the ASTM D130 standard [7]. The primary objective of ASTM D130 is to evaluate the relative corrosiveness of a petroleum product, particularly concerning the presence of sulfur compounds. However, in this study, pieces made from a bronze alloy (Table 1) with a heterogeneous microstructure were used. These included lead islands within a matrix primarily composed of Cu, along with other elements like Zn, Sn, Fe, and Ni. In this case, galvanic effects play a fundamental role in the corrosion mechanisms. For example, in the research by Bares et al., (2020) [11], it was observed that two brass alloys (Cu_63_Zn_37_ and Cu_85_Zn_15_) demonstrated greater resistance to corrosion in automotive gear oil (AGO) and automotive transmission fluid (ATF) compared to pure copper. Nonetheless, these alloys were still vulnerable to corrosion, showing a non-proportional dependence with the increase in the oil bath temperature (from 80 to 150 °C). In both alloys, the results were a consequence of the competition between mechanisms: the selective dissolution of Zn or Cu, and the formation of surface films detected by SEM-EDS (composed of S, P, and O in different percentages) [11].

The similar results between the tests conducted on MC-A and the mixture suggest that S-containing additives or sulfur compounds (identified by EDXRF, Figure 5a,c) cause a selective attack on the lead islands in the microstructure (as seen in Figure 7, Pb content in Table 4). In the case of MC-B, which also contains sulfur (confirmed by EDXRF, Figure 5), it is possible that the antioxidant additives or corrosion inhibitors were effective in protecting the surfaces. Another possibility is the effect of sulfur speciation in its corrosivity, as observed by [10,12,14] in copper corrosion tests. Indeed, Jayne, Shanklin, and Stachew (2002) [12] demonstrated that the presence of secondary ZDP produces severe corrosion of pure lead and negligible corrosion of pure copper, while primary and aromatic ZDP have the opposite effect.

Sources of sulfur include elemental sulfur present in base oils (group I, II, III), zinc dithiophosphates (ZDP), some antioxidants, extreme pressure additives, and sulfonates [12,22]. Although many sulfur compounds have been shown to be corrosive to copper in the literature [10,14], in this research the galvanic effect due to the microstructure of the bronze alloy likely prevailed. Lead is the second most significant metal in the alloy (Table 1), being less noble (more reactive) than copper in the galvanic series of elements.

Although the formulations of MC-A, MC-B, and SAE 10W-30 contain sulfur (as verified by EDXRF, Figure 5a,c), it was only detected on the bronze surfaces immersed in MC-A (Table 5). However, as mentioned previously, significant corrosive effects were observed only in MC-A and the mixture. In relation, Reid and Smith (2014) [10] concluded that the sulfur content (at. %, assessed by XPS) on the pure copper surfaces, was not proportional to the quantity of S-containing compounds, nor to the corrosiveness of the evaluated compound, but rather to the availability of elemental sulfur. Hence, the work of [10] demonstrated that the corrosiveness classification of ASTM D130 does not correlate proportionally with the sulfur concentration on the surface of pure copper. The findings from [10] could shed light on the corrosive effects of S-compounds in MC-A and the mixture, despite sulfur being detected only on the surface submerged in MC-A.

On the other hand, the corrosive effect of Cl-containing additives (content verified by EDXRF in MC-C and MC-D, Figure 5b) on the bronze alloy under study was observed. These additives showed selective corrosion of the matrix (mainly Cu, the major element), resulting in corrosive wear with the detachment of copper (MC-C) and copper + zinc (MC-D) from the surfaces. The corrosive process in both lubricants produced a layer of oxides and corrosion products that protruded from the surface, where chlorine was the sole additive element detected by SEM-EDS (Table 5). The non-detection of tin (in ppm, Table 4) by ICP-OES is likely attributable to its concentration being below the equipment’s lower detection limit [18]. Cl-containing additives have a greater affinity (and higher chemical reactivity) with metallic surfaces, hence their consideration as being more effective in forming a sacrificial layer compared to S-containing additives [16,22]. Despite the galvanic differences of the high-leaded tin bronze alloy, which theoretically favor the corrosion of other elements before copper (order of nobility or resistance to corrosion Cu > Sn > Pb > Zn), the chlorinated compounds showed a marked affinity with the matrix, as seen in the SEM-EDS maps of Figure 7b.

Any corrosion process in a liquid medium involves anodic (oxidation) and cathodic (reduction) reactions. The metals dissolves into metallic cations, a process that generates electrons consumed by cathodic reactions. In conditions with the presence of oxygen (in the air), cathodic reactions are based on a reduction in molecular oxygen. Possibly, the corrosion initiated at the interface in the MC-D tests (Figure 2d) was favored due to the physical proximity and availability of oxygen to be reduced in this region. This implies that anodic reactions began in the alloy in the early days of the test, promoted by highly reactive Cl-containing additives on the surfaces. Likely due to the rapid progression of corrosion, the process remained localized at the interface and propagated at a slower rate in the immersed region (Figure 2i,n).

On the other hand, the Cl-containing additives in the MC-C formulation showed a delayed reaction with the surfaces, leading to a general attack on the immersed surface (Figure 2m). The outside regions (non-immersed) of the parts tested in both products, MC-C and MC-D, also showed oxide formation and chlorine adsorption (Table 5). This indicates that the vapors generated during the test period also had a corrosive potential on the bronze alloy.

It is important to highlight that the chlorinated additives in MC-D, as well as those in SAE 10W-30, identified by EDXRF (Figure 5b,c), did not exhibit corrosive effects in the tests conducted with a mixture of both lubricants. This occurred despite the detection of chlorine on the surfaces immersed in the mixture (Table 5). As mentioned earlier, the mixture showed only selective dissolution of lead (as observed in the ppm content), a phenomenon not observed when MC-D was tested individually. This result suggests that the S-containing additives from the 10W-30 itself predominated in the reaction with the surfaces, leading to the formulation of three hypotheses: (i) A higher availability of sulfur compounds from the 10W-30, compared with the content of Cl-containing additives; (ii) a possible synergy with other additive compounds (Na, Ca: Table 5) that are not present in MC-D; (iii) selective protection of the alloy’s copper by the corrosion inhibitors in the formulated oil.

In field tests with motorcycles, the concentration of copper (in ppm) in the used oils remained within reference limits, with and without the addition of a metal conditioner in the crankcase oil [15]. These results align with those obtained in the present research, suggesting that the potential corrosive power of the chlorinated additives from the metal conditioner is negligible when mixed with a fully formulated engine oil. As observed by [12] in the corrosion of Cu and Pb and by [11] in the corrosion of brass, there is a synergy among additives that determines the corrosive power of the formulation. In this case, the synergy between the additives of SAE 10W-30 and the metal conditioner was dominated by the corrosiveness of the S-containing additives in the formulated oil.

### 4.3. Corrosiveness and Oil Partial Degradation Products

All lubricants exhibited minor changes in the infrared spectrum, such as the reduction in the intensity of the ester peak (Figure 4a) and the increase in absorption of the intermolecular O-H bond (Figure 4b). FTIR analyses of the metal surfaces revealed the adsorption of the O-H bond on parts immersed in MC-A, MC-C, MC-D, and the mixture (Figure 6). Both results indicate that the products of partial oil degradation were reactive with the bronze alloy and, consequently, became adsorbed.

Kalantar and Levin (2008) [9] studied how degradation compounds in oils, such as radicals, acids, and peroxides, can catalyze the corrosion of copper. The authors suggest that under aerated conditions, the pathway to Cu dissolution is as follows: The interaction of copper oxides or metallic copper with partial oxidation products of the oil (such as hydroperoxides ROOH and free radicals RO•) results in the dissolution of copper and/or the formation of CuO. As the concentration of soluble copper (Cu cationic) in the oxidized oil increases, a second stage begins in which these cations cause the catalytic oxidation of the oil [9].

In the experimental conditions established in this study, it is probable that only the initial phase outlined by Kalantar and Levin (2008) [9] was initiated. Notably, the absorbance in the 3600–3150 cm^−1^ region (Table 4) was observed to be higher in the oil that had been in contact with the bronze piece, compared to the control oil sample at 28 days. This was the case for the oils with corrosive effects: MC-A, MC-C, MC-D, and the mixture. This indicates that, although minimal, a catalytic effect of the alloy was present. This observation might account for the detection of O-H bonds in both the drained oils and on the surfaces, and their contribution to the formation of oxides. Nonetheless, the findings of this study revealed that the corrosivity of the products evaluated was predominantly dependent on the type of extreme pressure additive (S and Cl additives) used. It was also discerned that under the conditions tested, the partial degradation products of the oil played a minor role in the corrosion process.

## 5. Conclusions

This study evaluated the corrosive effects of four aftermarket metal conditioners on high-leaded tin bronze alloy, highlighting how S- and Cl-based additives interact with the alloy elements over 28 days at 80 °C. This research demonstrates that not all metal conditioners are inherently corrosive. This expands the current understanding of the impact of aftermarket additives on engine corrosion.

Notably, the results show that not all S-containing additives were corrosive, with selective lead corrosion noted after 14 days. Instead, Cl- containing additives were associated with pronounced copper corrosion and even zinc dissolution, particularly in tests conducted with pure metal conditioner alone.

Importantly, these corrosive effects were mitigated when conditioners were mixed with fully formulated oil, the standard usage for engine crankcases. This research offers foundational insights into metal conditioner and motor oil interactions with corrosion susceptible alloys, aiming to guide the development of safer, more effective aftermarket products.

## Figures and Tables

**Figure 1 materials-17-01326-f001:**
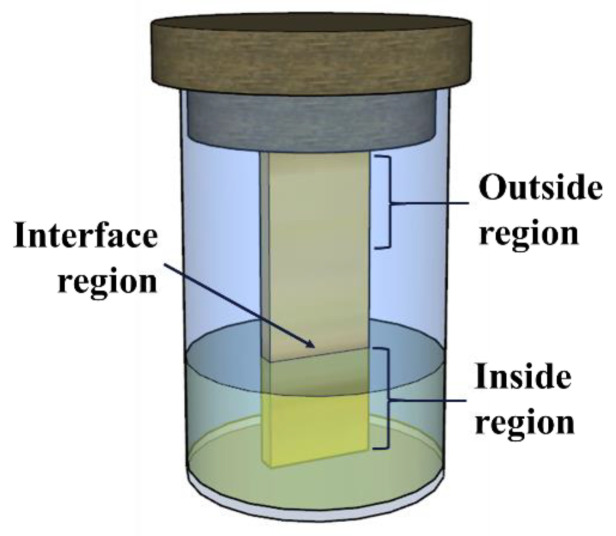
Schematic design of the immersion test.

**Figure 2 materials-17-01326-f002:**
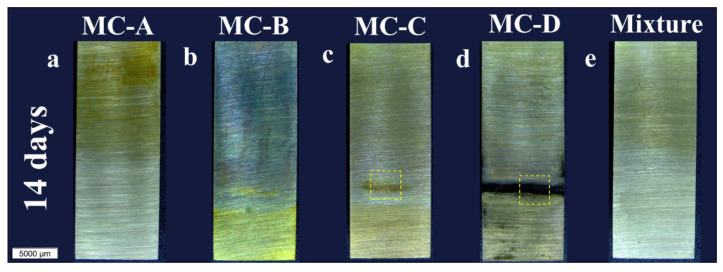
Stereoscopic macrograph of the end-of-test condition for 14, 21 and 28 days. Oil bath at 80 °C in MC-A (**a**,**f**,**k**), MC-B (**b**,**g**,**l**), MC-C (**c**,**h**,**m**), MC-D (**d**,**i**,**n**) and mixture (25:1) of SAE 10W-30 and MC-D (**e**,**j**,**o**). The colored rectangles represent the enlarged areas shown in Figure 3.

**Figure 3 materials-17-01326-f003:**
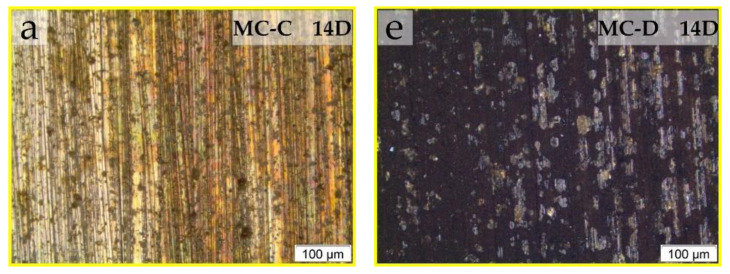
Optical micrographs enlargements of the end-of-test condition for 14, 21 and 28 days. Oil bath at 80 °C in MC-C (**a**–**d**) and MC-D (**e**–**h**). The magnification regions are identified in Figure 2.

**Figure 4 materials-17-01326-f004:**
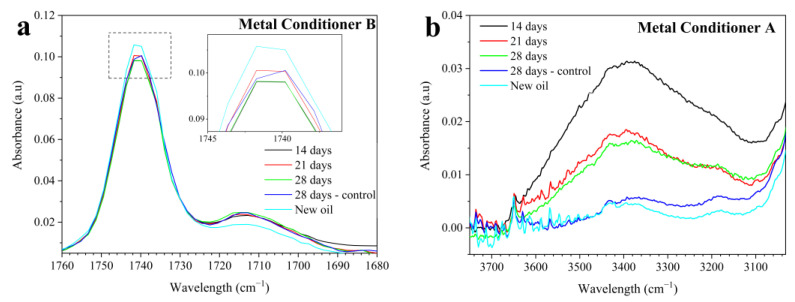
Representative infrared spectrum of (**a**) the oxidation region; (**b**) the OH absorption region.

**Figure 5 materials-17-01326-f005:**
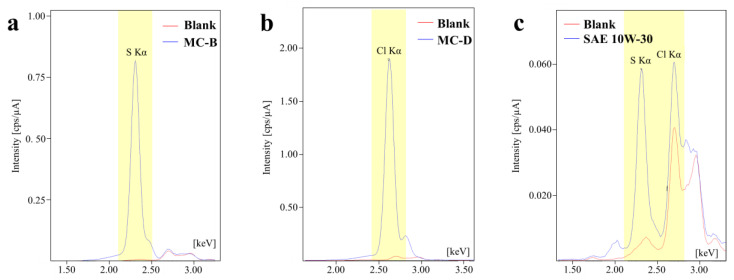
Energy Dispersive X-ray Fluorescence (EDXRF) spectra of (**a**) MC-B, (**b**) MC-D, and (**c**) SAE 10W-30 API SL JASO MA.

**Figure 6 materials-17-01326-f006:**
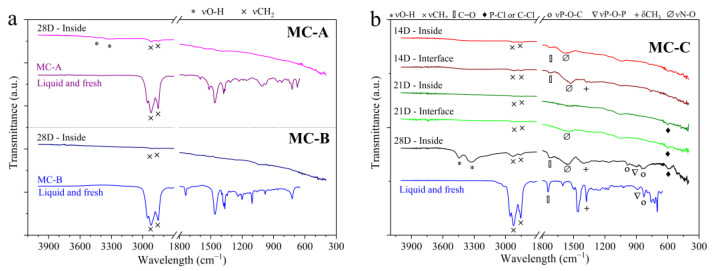
Surface characterization by FTIR of bronze pieces tested at different time intervals: (**a**) MC-A and MC-B; (**b**) MC-C; (**c**) MC-D; (**d**) the mixture. Band assignment based on [13,17].

**Figure 7 materials-17-01326-f007:**
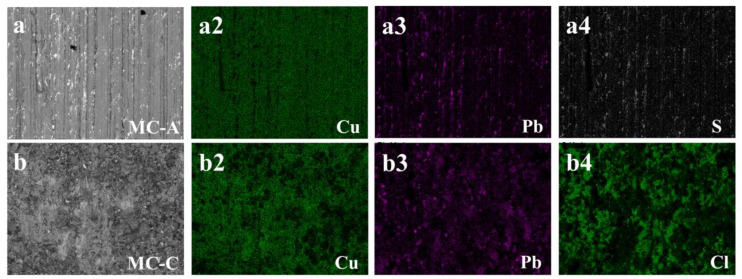
Elemental mapping images using SEM/EDS analysis of the end-of-test (28 days) surfaces immersed in (**a**–**a4**) MC-A and (**b**–**b4**) MC-C. The maps for copper (**a2**,**b2**), lead (**a3**,**b3**), sulfur (**a4**), and chlorine (**b4**) are respectively organized.

**Table 1 materials-17-01326-t001:** Chemical composition of high-leaded tin bronze alloy (SAE TM 23).

	Cu	Pb	Sn	Fe	Ni	Zn	O
Supplier (wt.%)	72	15	4	0.5	0.9	7.6	-
SEM-EDS (wt.%)	74.21	8.57	4.81	0.29	-	8.6	3.52

**Table 2 materials-17-01326-t002:** Summary of the commercial lubricants tested in this study. The information was taken from their technical datasheets.

	Metal Conditioner A (MC-A)	Metal Conditioner B (MC-B)	Metal Conditioner C (MC-C)	Metal Conditioner D (MC-D)	SAE 10W-30 API SL JASO MA (10W-30)
Kinematic Viscosity at 40 °C (cSt)	60–70	43–48 (ASTM D-445)	43–48 (ASTM D-2270)	43–48 (ASTM D-2270)	69–71 (ASTM D-445)
Kinematic Viscosity at 100 °C (cSt)	Not informed	7–9 (ASTM D-445)	Not informed	Not informed	10–12 (ASTM D-445)

**Table 3 materials-17-01326-t003:** Oxidation (abs/cm) and absorbance in the O-H bond region (abs/cm).

	New	14D	21D	28D	28D-Control
Oxidation (abs/cm) at region 1800–1670 cm^−1^					
MC-A	0.52	0.50	0.43	0.34	0.45
MC-B	2.43	2.56	2.49	2.47	2.54
MC-C	3.03	4.32	4.76	2.87	2.93
MC-D	29.93	29.85	30.26	30.54	29.89
Mixture	31.21	29.21	31.96	32.57	30.67
OH bond (abs/cm): region 3600–3150 cm^−1^					
MC-A	1.22	10.98	5.99	5.44	1.73
MC-B	0.08	6.84	2.15	1.27	2.69
MC-C	1.47	9.46	2.43	1.28	1.08
MC-D	1.23	1.79	7.52	5.15	3.62
Mixture	0.29	2.69	4.92	2.83	2.54

**Table 4 materials-17-01326-t004:** Element analysis (ppm) of end-of-test (28-days) oil drain for bronze pieces tested in MCs and mixture (ASTM D5185 [18]).

Lubricant	Condition	Cu	Pb	Zn	P	Ca	Mg	B	Na
MC-A	Fresh	0.50 ± 0.06	1.0 ± 1.6	2352 ± 140	5732 ± 200	7233 ± 506	9.2 ± 1.1	39.6 ± 10.3	13.4 ± 2.7
	14D	1.50 ± 0.17	31.5 ± 4.8	1516 ± 94	2547 ± 123	3014 ± 150	0	11.5 ± 2.3	1.0 ± 0.5
	21D	1.40 ± 0.16	39.1 ± 5.2	1712 ± 105	2951 ± 134	3525 ± 185	0	15.3 ± 4.0	2.3 ± 0.8
	28D	0	54.4 ± 5.7	2128 ± 127	3537 ± 150	4189 ± 235	0	16.6 ± 4.3	4.5 ± 1.3
	28D Control	0	0	1660 ± 102	2618 ± 125	3193 ± 161	0	10.6 ± 2.8	1.5 ± 0.6
MC-B	Fresh	0.50 ± 0.06	1.4 ± 1.8	31 ± 3	190 ± 27	17.1 ± 0.1	2.3 ± 0.3	0	4.9 ± 1.4
	14D	0	0	28 ± 3	106 ± 20	77.6 ± 0.9	0	0	0
	21D	0	0	36 ± 3	128 ± 22	98.0 ± 1.2	0	0	0
	28D	0	0	23 ± 2	122 ± 21	91.9 ± 1.1	0	0	1.0 ± 0.5
	28D Control	0	0	21 ± 2	77 ± 16	81.1 ± 0.9	0	0	0
MC-C	Fresh	0.50 ± 0.06	0.6 ± 1.4	0	6.0 ± 3.7	0	1.2 ± 0.2	0	4.2 ± 1.3
	14D	0	0	0	0	0	0	0	0
	21D	0	0	0	6.0 ± 3.7	0	0	0	0
	28D	3.80 ± 0.40	0	0	2.4 ± 2.2	0	0	0	0
	28D Control	0	0	0	5.8 ± 3.6	0	0	0	0
MC-D	Fresh	0	0	0	1480 ± 90	0	0	0	0
	14D	21.40 ± 1.95	0	0	659 ± 56	0	0	3.2 ± 0.8	0
	21D	32.50 ± 2.85	0	28 ± 3	720 ± 60	5.70 ± 0.02	0	0	0
	28D	2.70 ± 0.30	0	32 ± 3	725 ± 60	13.70 ± 0.08	0	2.1 ± 0.5	0
	28D Control	0	0	0	1159 ± 78	0	0	0	0
Mixture: SAE 10W-30 + MC-D	Fresh	0	0	880 ± 60	1020 ± 72	1750 ± 69	20.0 ± 2.1	455 ± 118	587 ± 33
	14D	0	8.5 ± 3.2	189 ± 15	371 ± 40	401 ± 9	0	89.1 ± 23.2	61.2 ± 7.4
	21D	0	24.4 ± 4.4	276 ± 21	538 ± 50	531 ± 13	0	126.5 ± 32.9	85.3 ± 9.2
	28D	0	23.3 ± 4.3	325 ± 24	588 ± 52	622 ± 16	0	148.6 ± 38.6	93.2 ± 9.8
	28D Control	0	0	401 ± 29	742 ± 60	811 ± 24	0	168.9 ± 43.9	103.6 ± 10.5

**Table 5 materials-17-01326-t005:** Semi-quantitative SEM-EDS analysis of sample surfaces at the end of the test period (28 days), showing the weight percent (wt.%) of elements that constitute the oil additives. The remaining percentage, required to reach the total of 100%, was distributed among the alloying elements.

Lubricant	Region	O	Na	S	Cl	Ca
MC-A	Inside	3.11	-	3.19	-	-
	Outside	3.30	-	2.50	0.15	-
MC-B	Inside	5.10	-	-	0.12	0.98
	Outside	4.27	-	-	-	-
MC-C	Inside	10.42	-	-	16.94	-
	Outside	11.47	-	-	3.25	-
MC-D	Inside	9.43	-	-	10.15	-
	Interface	15.61	-	-	19.69	-
	Outside	8.51	-	-	16.00	-
Mixture	Inside	16.43	14.63	-	8.89	0.50
	Outside	4.08	-	-	0.67	-

## Data Availability

The data that support this work and its results are not available to be shared because they are under confidentiality agreements. Access to the data can be requested through an official document.

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
