# Peer review of "The Corrosive Effects of Aftermarket Oil Additives on High-Leaded Tin Bronze Alloy"

_materials, 2024, doi:10.3390/ma17061326_

Round 1
Reviewer 1 Report
Comments and Suggestions for Authors
This is an excellent manuscript on corrosion of bronze alloy in various oil additives for up to 28 day submersion. Using many advanced instrument for surface analysis and chemical analysis, the authors show detail and meticulous measurements on the alloy samples. Besides the fact that most of reported data are consistent with the literature (ref [10] in particular), the current work shows new understanding of the corrosion process. The degree of corrosivity of the liquids are logically deduced based on the measurement. The analysis is sound and conclusion convincing. The results and methods should have significant impacts on corrosion of such alloys in the industry.
I have a couple of minor comments. In Table 2, why can't the author measure viscosity of the A, C and D? Standard viscometer measurement can be done quite routinely. In fact, the viscosity before and after the immersion is quite relevant and should be reported. A related quantity is the surface tension of liquids involved (before and after immersion). A small surface tension (or contact angle) will attract the liquid to climb up the sample over time, and may or may not change the spectroscopic measurement at the liquid level.
Author Response
Authors’ response:
We value your feedback and appreciate the time you dedicated to reading and reviewing our manuscript.
To address your query, the viscosity values listed in Table 2 are derived from the technical sheets of each product, as noted in the table's title. These values were included in the manuscript as part of a standard informational table on lubricant products. It's important to clarify that the authors did not conduct any viscosity measurements on the oils, neither before nor after immersion. The intent of Table 2 is merely to show that these are commercially available products with accessible technical sheets, which are not cited to preserve the confidentiality of the brands involved.
Viscosity is indeed an important factor, as the reviewer highlighted, because it is associated with the oil's degradation level and, to some extent, with the diffusion of corrosion products in their liquid state. It is noteworthy that the viscosities of additives MC-B, MC-C, and MC-D (at 40°C, according to Table 2) are within the same range, yet they lead to markedly different levels of degradation (as seen in FTIR: Table 3) and corrosivity (illustrated in Figure 2). This finding indicates that while viscosity is a significant aspect, it does not serve as the predominant parameter.
Thus, we, the authors, believe that evaluating oil degradation via Fourier-transform infrared spectroscopy (FTIR) yields a more precise and specific analysis. This technique distinctly identifies degradation caused by oxidation, sulfation, and nitration—predominant degradation modes in lubricants. Notably, the spectroscopic analysis revealed no alterations in the sulfation and nitration spectra, with changes observed exclusively in the oxidation domain, as elaborated in the “Results - Analysis of Lubricants” section. Furthermore, the FTIR assessments of the drained oils disclosed minimal oxidation levels (refer to Table 3). Literature highlights the potential for oil degradation products to exhibit significant corrosive properties, a point underscored in our introductory discussion. However, the findings from this study indicate that the corrosiveness imparted by partially degraded oil products is negligibly minimal.
Regarding the surface tension, we agree with the reviewer's observations: a lower tension will promote an increase of the oil in the part and, as a result, the interface line will rise. As the reviewer commented, this effect can affect the result of spectroscopic characterization at the initial interface (day zero, at the liquid level, Figure 1) compared to the interface after immersion times. However, the FTIR measurements at the interface were performed at the initial interface line, where for all parts there were already changes in the color of the coupons (Figure 2), indicating that there had already been interaction of the fluids with the surfaces.
Reviewer 2 Report
Comments and Suggestions for Authors
This study reports on the corrosion behavior of a high-leaded tin bronze alloy in four different aftermarket additives. The authors found a correlation between the depletion of alloy elements on the bronze coupon surfaces and the content of metals in the drained oils. However, there is uncertainty about the accuracy or consistence of a set of measurements. Further research and analysis will be needed to meet the standards of publication.
1. The reliability of ICP-AES quantitative results is questionable due to the lack of a precise description of solution preparation for ICP-AES analysis from drained oils. Despite a reference is provided (ASTM D5185 ref 15), it is not of easy access. Readers need to have a clear idea of the chemical nature of the solution to properly analyze the results. Authors must clarify whether the solid corrosion products in drained oils (L. 406-408) were partially or totally dissolved during solution preparation, since in some cases they remain as solids (L. 329-330). Thus, authors must provide a detailed description of the preparation method to obtain the solution for ICP-AES analysis. In addition, authors must provide the ICP-AES quantitative results (Table 4) with their corresponding standard deviation values, providing a clear measure of the variability in the data.
2. Quantitative SEM-EDS results with standard deviation values reported on Table 5 require a precise calibration of the instrument to accurately measure the elemental composition of the samples. Authors must provide a detailed description about calibration of the instrument.
3. For a better understanding of corrosion mechanisms, the chemical composition as well as the structure of corrosion layers on the surface of bronze coupons are of decisive importance. Authors must provide a clear identification of corrosion layers (L. 6-16-617) by analytical techniques such as XRD and SEM-EDS. However, ultrasonic cleaning after corrosion test may have eliminated partially or totally the solid corrosion products. Then, the results of the experiments are inconclusive because the reliability of measurements is questionable.
Comments on the Quality of English LanguageMinor adjustments of the English language are needed
Author Response
February 23, 2024
Authors’ response of Manuscript ID materials- 2863267 to Reviewer 2 Comments
This study reports on the corrosion behavior of a high-leaded tin bronze alloy in four different aftermarket additives. The authors found a correlation between the depletion of alloy elements on the bronze coupon surfaces and the content of metals in the drained oils. However, there is uncertainty about the accuracy or consistence of a set of measurements. Further research and analysis will be needed to meet the standards of publication.
Comment 1: The reliability of ICP-AES quantitative results is questionable due to the lack of a precise description of solution preparation for ICP-AES analysis from drained oils. Despite a reference is provided (ASTM D5185 ref 15), it is not of easy access. Readers need to have a clear idea of the chemical nature of the solution to properly analyze the results. Authors must clarify whether the solid corrosion products in drained oils (L. 406-408) were partially or totally dissolved during solution preparation, since in some cases they remain as solids (L. 329-330). Thus, authors must provide a detailed description of the preparation method to obtain the solution for ICP-AES analysis. In addition, authors must provide the ICP-AES quantitative results (Table 4) with their corresponding standard deviation values, providing a clear measure of the variability in the data.
Authors’ response:
The analyses were conducted in an external laboratory specializing in lubricant analysis, which maintains all calibration protocols and procedures according to standards such as ASTM D5185 and ASTM D4951. Hence, the authors believed it was unnecessary to provide detailed descriptions. Nonetheless, in response to the reviewer's suggestion, the procedure for the solution preparation provided by the external laboratory has been included. Now it reads:
“The elemental composition of the oils before and after immersion were monitored using Inductively Coupled Plasma Optical Emission Spectrometry (ICP-OES, Perki-nElmer, Optima 7300 DV). These analyses were carried out by a laboratory specialized in lubricant analysis, which follows the procedures of the ASTM D5185 standard [15]: the oil sample was diluted ten times by weight with analytical grade solvent using an ultrasonic homogenizer: isoparaffin in most cases, and xylene for those that were difficult to be correctly homogenized in isoparaffin. However, this technique does not allow for the determination of chlorine...”
The authors agree the use of the term "particulates" (L. 358, 406 – 409) was not appropriate, as it generally leads readers to assume these are particles that were not properly diluted during the preparation process or were not completely atomized in the ICP-OES. To prevent misunderstandings, this term has been removed and, in some instances, replaced with "concentration." Also, we included the standard deviation values in Table 4.
Comment 2: Quantitative SEM-EDS results with standard deviation values reported on Table 5 require a precise calibration of the instrument to accurately measure the elemental composition of the samples. Authors must provide a detailed description about calibration of the instrument.
Authors’ response:
EDS is an elemental chemical characterization technique that, as the reviewer points out, can become quantitative (with high accuracy and precision) when using certified calibration standards of the same chemical nature and composition as the samples being evaluated. However, when such standards are not used, the technique is considered semi-quantitative. The SEM-EDS equipment used (SEM: Zeiss EVO MA15; EDS: Oxford Instruments X-MAX 20) allows for both quantitative and semi-quantitative chemical characterizations. In both cases, it features an automatic standard matrix correction, named XPP matrix correction, which corrects for atomic number and absorption and fluorescence effects.
The semi-quantitative analyses (Table 5) were conducted using the "standardless" analysis method, as recommended by the equipment manufacturer:
- Measurement parameters, as beam current and accelerating voltage, were kept constant for all samples.
- A beam measurement on pure copper was performed before analyzing each bronze sample.
- To calculate the chemical composition, the pre-installed standardization database was used. The elements in the spectra were automatically identified, and the results were presented as the percentage by weight (wt. %).
In this study, we employed SEM-EDS characterization to provide insights into the elements present on the sample surface, their spatial distributions, and relative concentrations. We emphasized that the findings were analyzed and discussed from a semi-quantitative perspective, highlighting the detection of elements derived from lubricants adsorbed on the surfaces at relative concentrations. Moreover, in the revised manuscript, we included micrographs and their corresponding EDS maps (Figure 7) to illustrate that certain elements exhibit specific distributions within the microstructure. To avoid any misinterpretations, we have specifically emphasized in the revised manuscript that these results should be considered both semi-quantitative and qualitative.
In response to the reviewer's suggestions, we have provided a more detailed description of the SEM-EDS characterization procedures in the revised manuscript, as outlined below:
“Additionally, micrographs with higher magnifications and semi-quantitative chemical compositions were taken using a Scanning Electron Microscope equipped with Energy Dispersive X-ray Spectroscopy (SEM-EDS; SEM: Zeiss EVO MA15 / EDS: Oxford Instruments X-MAX 20). Before SEM-EDS characterization of bronze surfaces, the pieces were removed from vacuum and cleaned again in an ultrasonic bath with pure acetone.
The SEM-EDS semi-quantitative analyses were conducted using the standardless analysis method, as recommended by the equipment manufacturer. Measurement parameters were kept constant for all samples. A beam measurement on pure copper was performed before analyzing each bronze sample. To calculate the chemical composition, the pre-installed standardization database was used. The elements in the spectra were automatically identified, and the results were presented as the percentage by weight (wt. %). All elements identified through EDS mapping (performed at a magnification of 400x) were taken into account, with the exception of carbon, which was excluded due to its classification as a contaminant (base oil residue). Maps were acquired in the immersed and external region, indicated in Figure 1.”
Comment 3: For a better understanding of corrosion mechanisms, the chemical composition as well as the structure of corrosion layers on the surface of bronze coupons are of decisive importance. Authors must provide a clear identification of corrosion layers (L. 616-617) by analytical techniques such as XRD and SEM-EDS. However, ultrasonic cleaning after corrosion test may have eliminated partially or totally the solid corrosion products. Then, the results of the experiments are inconclusive because the reliability of measurements is questionable.
Authors’ response:
Complete or partial removal of contaminants from the surfaces to be analyzed by our SEM-EDS (SEM: Zeiss EVO MA15 / EDS: Oxford Instruments X-MAX 20) is a prerequisite for successful characterization. Liquid contaminants derived from carbon, such as oil residues, can contaminate the internal chamber of the equipment and potentially damage the detectors. Therefore, proper cleaning of the samples, especially when they contain oils/lubricants, is a critical prerequisite for employing the SEM-EDS technique. The cleaning protocols are also necessary for other elemental chemical analysis techniques, where high vacuum environments are used, such as XPS, XANES, SEM-WDS, as conducted in various studies characterizing films formed on metallic surfaces from lubricating oils:
https://www.sciencedirect.com/science/article/pii/S0301679X18305474
https://doi.org/10.1016/j.triboint.2020.106201
https://doi.org/10.1179/175158407X189293
https://www.sciencedirect.com/science/article/pii/S004316482100106X
https://www.sciencedirect.com/science/article/pii/S0301679X20303637
https://www.sciencedirect.com/science/article/pii/S0301679X20305156
https://link.springer.com/article/10.1007/s11249-006-9123-7
In the revised manuscript, we have incorporated micrographs and their corresponding chemical composition maps obtained through SEM-EDS (Figure 7). Figure 7 reveals that, despite thorough cleaning, remnants of corrosion products and elements from the additives persisted on the surfaces. These maps distinctly illustrate the relationship between the additives in the lubricants and their affinity for the alloy elements constituting the microstructure.
The discussion and conclusions have been improved in the revised manuscript to better correlate the data derived from a variety of characterization techniques, including analyses of both drained oils and bronze parts. Given the absence of prior studies on the corrosiveness of commercial aftermarket additives, according to our knowledge, our findings serve as an initial exploration into how metal conditioners and motor oil formulations interact with corrosion-susceptible metal alloys. This research lays the groundwork for developing safer and more efficacious aftermarket products.
Reviewer 3 Report
Comments and Suggestions for Authors
The paper reflects a simple study detailing the influence of the additives on the alloy performance. The conclusion is quite lenghty. I would rather suggest this to consider the current conclusion as discussion and add a summary of the work as conclusion.
There were certain instances where values represented did not really carry any significance. for example: 'The intensity 389 results (cps/µA) for the S-Kα line were: 10.1053 (MC-B) > 2.6267 (MC-A) > 0.6969 (SAE 390 10W-30) > 0.2292 (MC-D) > 0.1868 (MC-C)' - is higher value good or bad? The authors needs to mention this and add text to this
Total acid number is represented in abstract and conclusion but it is not reflected anywhere in the text
Author Response
February 23, 2024
Authors’ response of Manuscript ID materials- 2863267 to Reviewer 3 Comments
The paper reflects a simple study detailing the influence of the additives on the alloy performance. The conclusion is quite lenghty. I would rather suggest this to consider the current conclusion as discussion and add a summary of the work as conclusion.
Authors’ response: The authors appreciate your observation. The conclusions were improved to highlight the main contributions of the work. They are now as follows:
“This study evaluated the corrosive effects of four aftermarket metal conditioners on high-leaded tin bronze alloy, highlighting how S- and Cl-based additives interact with the alloy elements over 28 days at 80°C. This research demonstrates that not all metal conditioners are inherently corrosive. This expands the current understanding of the impact of aftermarket additives on engine corrosion.
Notably, the results show that not all S-containing additives were corrosive, with selective lead corrosion noted after 14 days. Instead, Cl-containing additives were associated with pronounced copper corrosion and even zinc dissolution, particularly in tests conducted with pure metal conditioner alone.
Importantly, these corrosive effects were mitigated when conditioners were mixed with fully formulated oil, the standard usage for engine crankcases. This research offers foundational insights into metal conditioner and motor oil interactions with corrosion susceptible alloys, aiming to guide the development of safer, more effective aftermarket products.”
There were certain instances where values represented did not really carry any significance. for example: 'The intensity 389 results (cps/µA) for the S-Kα line were: 10.1053 (MC-B) > 2.6267 (MC-A) > 0.6969 (SAE 390 10W-30) > 0.2292 (MC-D) > 0.1868 (MC-C)' - is higher value good or bad? The authors needs to mention this and add text to this.
Authors’ response:
Intensity values are directly correlated with the concentration of the element within the fluid. This explanation was incorporated into the paragraph to enhance the reader's comprehension of its significance. Additionally, intensity values are included in the text to demonstrate that all samples were analyzed, although only some are shown in Figure 5. However, as mentioned in that section, these values must be interpreted with caution, since the presence of Cl and S was determined qualitatively through EDXRF.
Total acid number is represented in abstract and conclusion but it is not reflected anywhere in the text.
Authors’ response:
TAN measurements were not conducted in this research. The term was only mentioned in the literature review presented in the Introduction.
Reviewer 4 Report
Comments and Suggestions for Authors
Comment 1: What is the specific importance of bronze alloy in industries?
Comment 2: Qualitative informations are missing in abstract. Abstract should be concise and the authors need to improve with more specific short results.
Comment 3: All acronyms should be introduced at first place of appearance in the text. For example FTIR, SEM,... in the abstract section.
Comment 4: Quality of Figures 4 and 5 should be improved.
Comment 5: The novelty of the work should be established.
Comment 6: Adsorption mechanism should be added.
Comment 7: The introduction section should be modified though citing recent references related studies. Also, the following references should be included in the introduction part.
https://doi.org/10.1016/j.molstruc.2021.131014
https://doi.org/10.1016/j.cplett.2021.139081
Comment 8: Conclusion is trivial, Conclusion should be revised and improved.
Comment 9: Level of English is good however in a few places some syntax errors are present. At some places two or more words joined together that should be corrected.
Comments on the Quality of English LanguageSee comments
Author Response
February 23, 2024
Authors’ response of Manuscript ID materials- 2863267 to Reviewer 4 Comments
Comment 1: What is the specific importance of bronze alloy in industries?
Authors’ response: The characteristics and applications of these alloys are found in the section "2. Materials and Methods". Below, we provide the paragraph as it appears in the manuscript:
“The microstructure consists of islands or globules of lead within a matrix of copper, zinc, and tin, which are the principal elements of the alloy. This chemical composition and microstructure are responsible for its properties, such as high corrosion resistance (among bronze alloys), excellent anti-friction quality, optimal machinability, and formability. Owing to these properties, High-leaded Tin Bronze Alloys are preferred materials for manufacturing various components such as valves, bushings, bearings, sleeves, crowns, rings, and hydraulic materials.”
Comment 2: Qualitative informations are missing in abstract. Abstract should be concise and the authors need to improve with more specific short results.
Authors’ response: The abstract was improved according to reviewer’s suggestions:
“Aftermarket additives are used to enhance the performance of internal combustion engines in specific aspects such as reducing wear, increasing power, and improving fuel economy. Despite their advantages, they can sometimes cause corrosion-related problems. This research evaluated the corrosiveness of four aftermarket additives on the corrosion of a high-leaded tin bronze alloy over 28 days at 80 °C in immersion tests. Among the evaluated products, three showed corrosive effects ranging from intermediate to severe. Notably, the visual appearance of the surfaces often did not indicate the underlying corrosive damage. Therefore, the assessment of corrosiveness was based on chemical characterizations conducted on both the drained oils and the bronze surfaces. The study found minimal oil degradation under the testing conditions, indicating that the primary cause of corrosion was the interaction between the specific additives and the metal elements of the alloy, rather than oil degradation itself. A direct correlation was observed between the dissolution of lead and copper and the adsorption of S and Cl-containing additives on the surfaces, respectively. The corrosive impact of Cl-containing additives in aftermarket formulations was significantly reduced when mixed with engine oil SAE 10W-30 (at a 25:1 ratio), suggesting a mitigated effect in combined formulations, which is the recommended usage for engines.”
Comment 3: All acronyms should be introduced at first place of appearance in the text. For example FTIR, SEM,... in the abstract section.
Authors’ response: Indeed, we did not introduce the acronyms the first time they appeared in the text. This error was resolved by improving the abstract to make it more concise and objective.
Comment 4: Quality of Figures 4 and 5 should be improved.
Authors’ response: The authors appreciate your observation, the quality was improved.
Comment 5: The novelty of the work should be established.
Authors’ response: The novelty of the work was established in the last paragraph of the introduction section. Below, we provide the paragraph as it appears in the manuscript:
“Previous research indicates that analyzing the corrosiveness of oils and their compounds is a multi-component and complex issue. To the best of the authors' knowledge, there have been no studies focused on the corrosiveness of commercial aftermarket additives to date. Consequently, this study investigates the corrosive effects of four commercial aftermarket additives, specifically metal conditioners [12,13], on a bronze alloy. Utilizing a comprehensive approach, the research employs many surface characterization techniques, including optical microscopy, FTIR, and SEM-EDS, to analyze corrosion. Additionally, the study examines degradation, metal content, and additive consumption in drained oils using FTIR and ICP-OES, and detects chlorine and sulfur in additives with Energy-Dispersive X-ray Fluorescence Spectrometry (EDXRF). Together, these complementary techniques enabled the determination of the corrosiveness level of each additive.”
Comment 6: Adsorption mechanism should be added.
Authors’ response: A brief description of the adsorption mechanisms was added in the discussion section as shown below:
“Corrosiveness and oil additives relationship.
The adsorption of lubricant molecules on metal surfaces occurs primarily through two mechanisms: physical and chemical [19, 20]. The former, or physisorption, takes place when the interaction between the adsorbed molecule and the surface is weak, resulting from secondary chemical bonds. In contrast, chemical adsorption, or chemisorption, results from primary chemical bonds such as ionic or covalent bonds. Due to the temperature of the oil bath (80 °C) and the absence of tribological contact, it is expected that only the physical adsorption of additives onto the virgin surfaces occurred [9,10,19,20]. This would explain why only a few additive elements…”
Comment 7: The introduction section should be modified though citing recent references related studies. Also, the following references should be included in the introduction part.
Journal of Materials and Environmental Science, 8 (2017) 4349 – 4361. 10.26872/jmes.2017.8.12.458
Journal of Materials and Environmental Science, 8 (2017) 603 – 610.
https://doi.org/10.1016/j.molstruc.2021.131014
https://doi.org/10.1016/j.cplett.2021.139081
Authors’ response: Research on the corrosiveness of additives in lubricating oils towards copper-based alloys is limited. In fact, only a few sources, specifically the references [5, 6, 8, 9], address this topic. No scientific study or research from a reliable source has been identified that examines the corrosiveness of aftermarket additives. The other references mentioned in the introduction are relevant as they specifically focus on the corrosive effect of pure chlorinated and sulfured compounds, which differ from those found in lubricating oil formulations, on pure copper or lead. The references suggested by the reviewer were incorporated into the discussion section on types of adsorption, both chemical and physical.
Comment 8: Conclusion is trivial, Conclusion should be revised and improved.
Authors’ response: The authors appreciate your observation. The conclusions were improved to highlight the main contributions of the work. They are now as follows:
“This study evaluated the corrosive effects of four aftermarket metal conditioners on high-leaded tin bronze alloy, highlighting how S- and Cl-based additives interact with the alloy elements over 28 days at 80°C. This research demonstrates that not all metal conditioners are inherently corrosive. This expands the current understanding of the impact of aftermarket additives on engine corrosion.
Notably, the results show that not all S-containing additives were corrosive, with selective lead corrosion noted after 14 days. Instead, Cl- containing additives were associated with pronounced copper corrosion and even zinc dissolution, particularly in tests conducted with pure metal conditioner alone.
Importantly, these corrosive effects were mitigated when conditioners were mixed with fully formulated oil, the standard usage for engine crankcases. This research offers foundational insights into metal conditioner and motor oil interactions with corrosion susceptible alloys, aiming to guide the development of safer, more effective aftermarket products.”
Comment 9: Level of English is good however in a few places some syntax errors are present. At some places two or more words joined together that should be corrected.
Authors’ response: We appreciate your comment. We have improved the English throughout the manuscript.
Round 2
Reviewer 2 Report
Comments and Suggestions for Authors
After revision, the quality of manuscript has been improved. However, the authors did not answer some of my questions and concerns correctly. The manuscript still contains fundamental flaws in its argumentation, which need to be addressed before it can be published.
The authors report EDS semi-quantitative analysis results with standard deviation values on Table 5, which is still confused. In a EDS semi-quantitative analysis, the results are usually of some deviation from the actual situation. Should this not be the case, what precisely is the interpretation of standard deviation values on Table 5? Authors must explain it.
All discussions are based on the data from used oils and the resulting surface of samples upon ultrasonic cleaning, despite the authors stated the presence of corrosion layers on the surface (L. 585-586; 590-591). The ultrasonic cleaning after corrosion test may have eliminated partially or totally the solid corrosion products. Then, the reliability of discussions is questionable due to the inconclusive nature of results. Accordingly, authors must provide a clear identification of removed corrosion layers by ultrasonic cleaning and include the results on discussions of corrosion mechanisms.
Author Response
Our responses are attached

Reviewer 4 Report
Comments and Suggestions for Authors
Accept
Author Response
We value your feedback and appreciate the time you dedicated to reading and reviewing our manuscript.
Round 3
Reviewer 2 Report
Comments and Suggestions for Authors
The authors reply has been deemed acceptable and the manuscript has been accordingly improved in this version. Then, I would recommend the manuscript for publication in Materials.